# A single resistance factor to solve vineyard degeneration due to grapevine fanleaf virus

Samia Djennane[1,2], Emilce Prado[1,2], Vincent Dumas[1,2], Gérard Demangeat[1,2], Sophie Gersch[1], Anne Alais[1], Claude Gertz[1], Monique Beuve[1], Olivier Lemaire[1] & Didier Merdinoglu [1✉]

Grapevine fanleaf disease, caused by grapevine fanleaf virus (GFLV), transmitted by the soil-borne nematode *Xiphinema index*, provokes severe symptoms and economic losses, threatening vineyards worldwide. As no effective solution exists so far to control grapevine fanleaf disease in an environmentally friendly way, we investigated the presence of resistance to GFLV in grapevine genetic resources. We discovered that the Riesling variety displays resistance to GFLV, although it is susceptible to *X. index*. This resistance is determined by a single recessive factor located on grapevine chromosome 1, which we have named *rgflv1*. The discovery of *rgflv1* paves the way for the first effective and environmentally friendly solution to control grapevine fanleaf disease through the development of new GFLV-resistant grapevine rootstocks, which was hitherto an unthinkable prospect. Moreover, *rgflv1* is putatively distinct from the virus susceptibility factors already described in plants.

[1] INRAE, Université de Strasbourg, UMR SVQV, Colmar, France. [2]These authors contributed equally: Samia Djennane, Emilce Prado, Vincent Dumas, Gérard Demangeat. ✉email: didier.merdinoglu@inrae.fr

G rapevine is one of the most important perennial crops globally due to its impact on the economy, shaping landscape and cultural identities[1–4]. Nevertheless, a number of viruses affect vineyards all over the world and negatively impact berry quality, plant growth and yield, even leading to the death of chronically infected plants[5–7]. Apart from prophylactic measures based upon the determination of the sanitary status of the vines, there is no method to control these viral diseases[8]. Grapevine fanleaf disease is the most detrimental of these diseases. The main causative agent of grapevine fanleaf disease is grapevine fanleaf virus (GFLV), a soil-borne *Nepovirus* transmitted by the dagger nematode, *Xiphinema index*[9]. Grapevine fanleaf disease causes many symptoms, including leaf deformation, yellowing, mosaicking, vein banding, abnormal branching and shortened internodes, and leads to irregularly ripening clusters and relatively small berries[10]. Yield losses can reach 77% in the most severe cases[11]. The resulting grapevine degeneration threatens vineyards worldwide[10,12], and leads to economic losses estimated at US $16,600 per hectare[8].

Many genetic engineering strategies have been proposed to control grapevine fanleaf disease. Pathogen-derived resistance has been intensely exploited, and led to the production of transgenic grapevine plants expressing the GFLV capsid protein[13–19]. Approaches based on gene silencing, through the use of amiRNAs or hairpins, have also been attempted[20–22]. However, regardless of the strategy employed, no effective resistance to GFLV has been observed in transgenic grapevines. More recently, immunity towards GFLV has been demonstrated in nanobody-expressing rootstocks[23]. However, their experimental use in field trials and their commercial release have not been accepted so far by society, especially in Europe[8]. Methods based on biological control have also been explored. The nematicidal properties of several plant species and their use in fallow fields between two successive vine crops have been assessed. Some of these plants have shown an antagonistic effect on *X. index* multiplication in controlled conditions, thus allowing a reduction in soil inoculum[24]; nevertheless, experiments in vineyards are still ongoing. Cross protection, based on the use of mild viral strains as a protectant to limit the expression of a subsequent challenging virus, has been tested against GFLV. Its application in vineyards did not achieve the expected results, with a yield loss having been measured along with the delay in the infection of vines by the GFLV natural strains[25].

Natural resistance to grapevine fanleaf disease has also been explored. Immunity to the feeding of the vector nematode, *X. index*, correlated with an absence of GFLV infection has been found in muscadine grapes[26]. This resistance has been shown to display a complex genetic determinism involving at least three genomic regions[27]. Breeding programmes aimed at introducing muscadine resistance into rootstocks were undertaken[28,29], but the resulting new rootstock varieties had limitations. Even if they were able to delay the contamination of new plantations, the rootstocks tested positive for GFLV in infested vineyards and displayed some agronomical deficiencies[30,31]. Several works have been implemented to find a source of natural resistance to GFLV[32–34]. In the most complete screening study, a large number of accessions sampled in the different species of the *Vitaceae* family were inoculated using GFLV-infected green cuttings, but none of the tested accessions were found to be resistant to GFLV[32].

As recessive genetic determinisms have often been described for plant–virus resistance[35,36], the challenges of finding a source of natural resistance to GFLV may be explained by the high rate of heterozygosity in grapevine[37]. In this study, we aimed to explore the presence of recessive resistance to GFLV in grapevine genetic resources. We tested the resistance of the progenies of

self-fertilisation of various grapevine varieties and species to grapevine fanleaf disease. We discovered that the Riesling cultivar displays resistance to GFLV determined by a single locus. To our knowledge, this is the first and only instance of resistance to GFLV identified in grapevine so far.

## Results

**Riesling and its self-progeny display a unique resistance to grapevine fanleaf disease.** Eleven progenies derived from self-fertilisation (S1) of *Vitis vinifera* varieties or *Vitis* species were screened for the presence of resistance to grapevine fanleaf disease. The S1 seedlings were evaluated in containers filled with soil naturally contaminated by viruliferous nematodes. The plants were tested for GFLV presence by double-antibody sandwich enzyme-linked immunosorbent assay (DAS-ELISA) 2 years after planting. The plants resulting from self-fertilisation of the Riesling variety were the only ones that displayed a resistant phenotype. GFLV was not detected in any of these plants, while the S1 generations derived from the other accessions were heavily infected, at rates of 60–100% (Table 1). The homogeneity of the resistant phenotype observed in the S1 Riesling progeny suggests that the Riesling variety carries resistance to grapevine fanleaf disease in a homozygous state.

In addition, the resistance of Riesling and four well-known reference varieties (SO4, Kober 5BB, VRH8771 and Gewurztraminer) when challenged with GFLV-viruliferous nematodes in controlled conditions was compared. The presence of GFLV in root samples was tested by DAS-ELISA 8 weeks post inoculation. Of the plants with genotypes SO4, Kober 5BB and Gewurztraminer, one hundred percent were GFLV positive. In contrast, the rate of GFLV-infected roots was very low for VRH8771 and Riesling, at 15% and 13%, respectively (Table 2). DAS-ELISA was performed on leaves 18 months post inoculation, including a dormant period during the winter. As observed in the roots, all the susceptible genotypes were GFLV positive. The Riesling plants remained uninfected by GFLV, while 6 plants out of 11 biological replicates tested GFLV positive for the VRH8771 genotype

**Table 1 Infection rate of 11 S1 progenies derived from grapevine varieties or *Vitis* species, and the susceptible controls (sc).**

| Genotype | Plant material origin | Number of analysed plants | % of GFLV positive plants |
|---|---|---|---|
| Cabernet Sauvignon (sc) | Green cuttings | 27 | 100 |
| 40024 (sc) | Seed sowing | 8 | 100 |
| S1 Cesar | Seed sowing | 56 | 84 |
| S1 Cot | Seed sowing | 9 | 89 |
| S1 Mauzac | Seed sowing | 15 | 60 |
| S1 Merlot | Seed sowing | 26 | 70 |
| S1 Mondeuse | Seed sowing | 14 | 85 |
| S1 Riesling | Seed sowing | 8 | 0 |
| S1 Tempranillo | Seed sowing | 6 | 100 |
| S1 Uburebekur | Seed sowing | 6 | 100 |
| S1 *Vitis berlandieri* | Seed sowing | 4 | 100 |
| S1 *Vitis lincecumii* | Seed sowing | 15 | 100 |
| S1 *Vitis monticola* large bell | Seed sowing | 22 | 91 |

The percentage of grapevines infected by GFLV is revealed by DAS-ELISA on leaves, following *X. index*-mediated transmission performed through 24 months of greenhouse cultivation in containers filled with soil naturally contaminated by viruliferous nematodes.

**Table 2 Infection tests of five genotypes of grapevine varieties or rootstocks.**

| Genotype | DAS-ELISA on roots[a] | | DAS-ELISA on leaves[b] | |
| --- | --- | --- | --- | --- |
| | GFLV-positive plants/tested plants | % of positive plants | GFLV-positive plants/tested plants | % of positive plants |
| SO4 | 15/15 | 100 | 15/15 | 100 |
| Kober 5BB | 10/10 | 100 | 10/10 | 100 |
| VRH8771 | 2/13 | 15 | 6/11 | 55 |
| Gewurztraminer 643 | 15/15 | 100 | 15/15 | 100 |
| Riesling 49 | 1/8 | 13 | 0/8 | 0 |

The number of grapevines infected by GFLV was revealed by DAS-ELISA, following *X. index*-mediated transmission.
[a]Results of DAS-ELISA performed on roots after a period of 8 weeks of contact with GFLV-viruliferous *X. index*.
[b]Results of DAS-ELISA performed on leaves after 18 months of cultivation in a greenhouse.

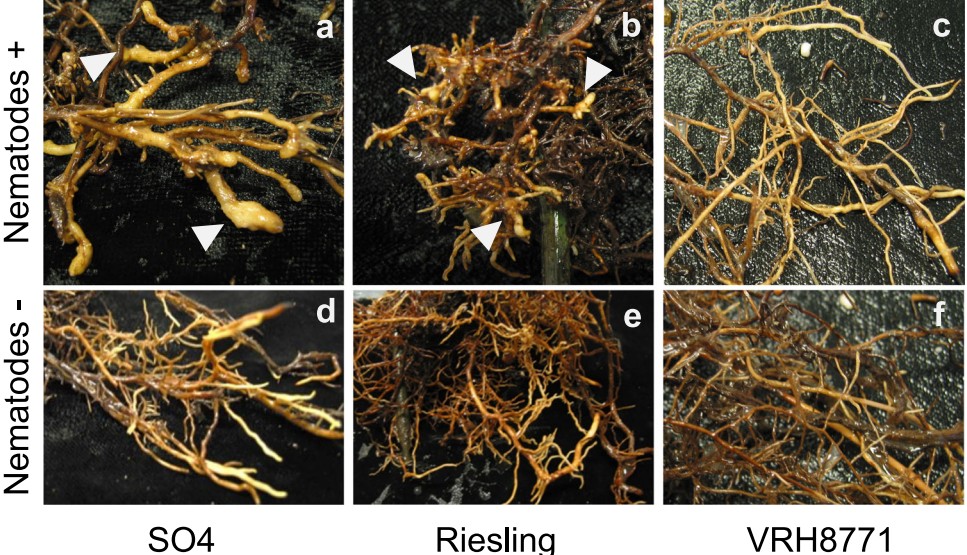

**Fig. 1 Impact of *X. index* feeding on root tips.** Root systems of the nematode-susceptible rootstock (SO4; **a**, **d**), the Riesling cultivar (**b**, **e**) and a nematode-resistant genotype (VRH8771; **c**, **f**) that were exposed to the dagger nematode, *X. index*, for 8 weeks (**a–c**) or cultivated in soil free of nematodes (**d–f**). The roots of the SO4 and Riesling varieties show terminal galling and swelling (white arrows) caused by *X. index*. Conversely, no root deformation was observed in the VRH8771 genotype or in the nematode-free controls.

(Table 2). These results, obtained in controlled *X. index*-mediated transmission assays, confirmed that the resistance to grapevine fanleaf disease previously observed in Riesling S1 progeny is also expressed in the Riesling variety itself.

**Riesling is resistant to GFLV but susceptible to *X. index*.** The fanleaf disease pathosystem is composed of a causative viral agent, GFLV, and a specific nematode vector, *X. index*. Controlled nematode-mediated transmission assays allowed us to observe that, except the VRH8771 genotype, which is already known to be resistant to the nematode vector, all the tested genotypes, including Riesling, displayed terminal galling and swelling on roots caused by *X. index* feeding (Fig. 1). This suggests that the fanleaf disease resistance observed in Riesling is directed against GFLV, but not against *X. index*. To confirm this hypothesis, we assessed the host suitability of Riesling to the vector nematode (*X. index*). For this, we measured the reproduction factor (RF) of the dagger nematode on Riesling roots, and compared this value with two reference genotypes: a nematode-susceptible rootstock (SO4) and a nematode-resistant genotype (VRH8771).

After 12 months of contact between the root systems and the nematodes, the average RF was 1.0 for the VRH8771 genotype, which is in accordance with its expected nematode-resistance

behaviour. In contrast, SO4 displayed an average RF of 8.2, as expected for a nematode-susceptible genotype. The RF observed for Riesling (6.7) is similar to that of SO4, which indicates that *X. index* normally multiplies when in contact with a Riesling root system (Fig. 2a). As observed previously, the roots of Riesling and SO4 genotypes showed many terminal galls and swelling, while the roots of the VRH8771 genotype remained unaffected. After 1 year of exposure to *X. index*, Riesling and SO4 root development was reduced compared to that of unexposed plants cultivated in the same conditions. The presence of GFLV was monitored in plant roots at the end of the experiment. All the SO4 plants were GFLV positive according to DAS-ELISA (mean $OD_{405\,nm} = 0.3$), while the Riesling and VRH8771 plants displayed a mean $OD_{405\,nm}$ of 0.007, below the GFLV detection threshold (Fig. 2b). Taken together, these results confirm that the resistance to grapevine fanleaf disease discovered in Riesling is directed against the causative agent (GFLV) and not against the vector (*X. index*).

**Riesling resistance to GFLV is governed by a single recessive factor.** To decipher the genetic basis of the resistance to GFLV discovered in Riesling, a set of 14 individuals derived from a cross between Riesling and the susceptible cultivar Gewurztraminer (Rs × Gw) was used (Fig. 3). Five to six green cuttings of each

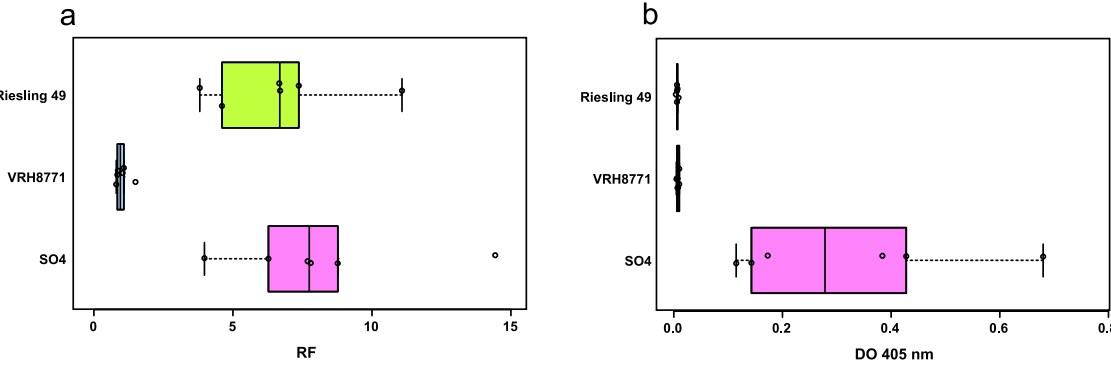

**Fig. 2 Characterisation of the nature of the Riesling resistance to grapevine fanleaf disease.** Evaluation of nematode multiplication (**a**) and DAS-ELISA (**b**) performed on roots of three grapevine genotypes. The centre lines show the medians, the box limits indicate the 25th and 75th percentiles, the whiskers extend to 1.5 times the interquartile range from the 25th and 75th percentiles, and the data points are plotted as open circles. $n = 6$ sample points.

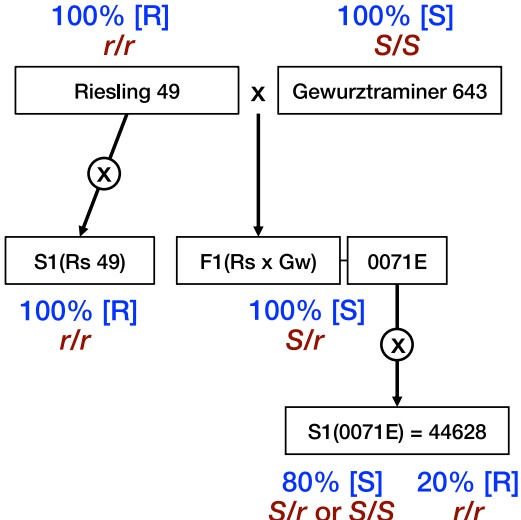

**Fig. 3 Overview of the pedigrees of the populations used for genetic analysis.** The resistance phenotypes observed from leaves in each population are in blue in the square brackets (R indicates resistant and S indicates susceptible), and the corresponding presumed genotypes are in dark red in italics ($r = rgflv1$).

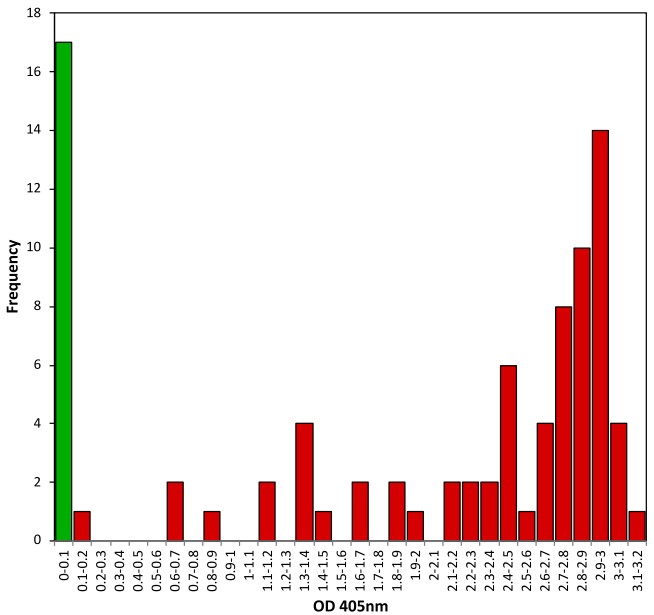

**Fig. 4 Distribution of ELISA values in the 44628 progeny (87 individuals).** The resistant plants (17 individuals) are in green and display a max_ELISA ranging from 0.004 to 0.011 (Supplementary Data 1), i.e., lower than the detection threshold (0.012). The susceptible plants (70 individuals) are in red.

genotype were infected through 36 months of cultivation in containers filled with soil naturally contaminated by viruliferous nematodes. One hundred percent of the Kober 5BB control plants were infected by GFLV. Similarly, all the Rs × Gw genotype plants were shown to be susceptible to GFLV, with 83–100% of the tested cuttings infected by the virus (Supplementary Table 1). This result strongly suggests that Riesling GFLV resistance is governed by a recessive factor, while the Gewurztraminer variety is homozygous for susceptibility.

To test this hypothesis, we developed a new progeny (named 44628) from one generation of self-fertilisation (S1) of Rs × Gw individual 0071E (Fig. 3). DAS-ELISA was performed to evaluate the resistance of 113 individuals, which were cultivated for 4 years in a container filled with soil naturally contaminated by viruliferous nematodes. The randomly distributed susceptible control plants were all contaminated by GFLV, indicating that the container was evenly infested (Supplementary Fig. 1). For each '44628' individual, the highest $OD_{405\,nm}$ value (max_ELISA) observed over the 4 years of the experiment was considered the best indicator of the resistance phenotype. Only 87 individuals out of 113 received max_ELISA scores, because 26 individuals

died before the third year of the experiment. Segregation of the 44628 S1 population exhibited a bimodal distribution of the values, which ranged from 0.004 to 3.127 (Fig. 4 and Supplementary Data 1). Seventeen plants displayed a max_ELISA lower than the detection threshold (0.012), and were thus considered resistant to GFLV. The 70 remaining plants exhibited max_ELISA scores higher than the threshold (ranging from 0.127 to 3.127), and were therefore classified as susceptible to GFLV. A chi-square goodness-of-fit test showed that the observed distribution is not significantly different from the expected 3:1 Mendelian segregation ($\chi^2 = 1.3831$, $p$ value = 0.2396). No structure was observed between the spatial distribution of the resistant and susceptible '44628' plants (Supplementary Fig. 1).

Altogether, the phenotypes observed in the Riesling and Gewurztraminer varieties, and the Riesling S1, Rs × Gw and 44628 progenies indicate that Riesling resistance to GFLV is governed by a single recessive genetic factor. While Gewurztraminer carries an allele for susceptibility in the homozygous

state, Riesling is, at the same locus, homozygous for this new resistance factor, which we name *rgflv1* for 'resistance to grapevine fanleaf virus 1' (Fig. 3).

***rgflv1* is located on grapevine chromosome 1**. To identify the chromosomal location of *rgflv1*, 11 resistant and 11 susceptible individuals from 44,628 progeny were selected to detect intergroups polymorphisms using a set of SSR (simple sequence repeats) markers well distributed throughout the genome. All 101 tested SSR markers segregated in both the resistant and susceptible groups of 44628 progeny or, at least, in one of them. A chi-square two sample test showed that four markers (VMC4f8, VVIq35, VVIc72 and VVCS1H024F14R1-1), all of which were located on chromosome 1 in a 3.3 Mb region, displayed a significantly different distribution between the two groups (*p* value < 0.0001; Supplementary Fig. 2 and Supplementary Data 2). In particular, the segregation of VVIq35 showed the strongest association with resistance: all the resistant plants (11/11) were homozygous for allele 389, whereas all the susceptible plants were either homozygous for allele 385 (4/11) or heterozygous (7/11).

All 87 phenotyped individuals in the population 44,628 were then used to map the *rgflv1* locus on chromosome 1. In addition to the four markers covering the region of interest, six new SSR markers were developed from *V. vinifera*'s reference genome (PN40024 12Xv2) to densify markers around *rgflv1* (Supplementary Table 2 and Supplementary Data 3). Segregation for resistance to GFLV was encoded as a qualitative trait; based on our previous results, susceptibility was considered dominant, and resistance was considered recessive. For both the markers and resistance, no significant difference was detected between the observed and expected Mendelian ratios. All ten SSR markers were mapped in a 14.4 cM long segment. *rgflv1* was located in a 5.7 cM interval between markers VMC4f8 and Chr1_1535, which represents a physical distance of ~1.1 Mb (Fig. 5). Altogether, these results allow us to conclude that Riesling resistance to GFLV is governed by a single recessive locus, *rgflv1*, located on the upper arm of grapevine chromosome 1.

## Discussion

Our work describes the first and, to date, the only instance of efficient genetic resistance to grapevine fanleaf disease discovered in cultivated grapevines. Indeed, we have found that Riesling, an elite variety highly prized by wine consumers around the world, displays a unique resistance to grapevine fanleaf disease. Although Riesling is broadly cultivated in many wine-producing countries, hypotheses can be put forward to explain why the resistance discovered in this study has not been previously observed and revealed in vineyards. Grapevine varieties are generally grown by grafting them onto rootstocks, all of which are susceptible to grapevine fanleaf disease. If the resistance is only effective at the root level, then grafted plants, which do not grow from their own roots, will not express it. Furthermore, if cultivated on contaminated soil, a rootstock multiplies GFLV and can thus transmit the virus to the grafted variety it supports through a high dose of inoculum, which is difficult to control, even in the case of resistance. High viral load has already been pointed out by Lahogue and Boulard[32] to explain the challenges of finding natural resistance to GFLV. Finally, the heterogeneity of plots, in terms of infection by the fanleaf disease pathosystem, and the diversity of symptoms caused by the disease may have been additional obstacles to the identification of Riesling resistance in the vineyard.

The absence of virus simultaneous to the multiplication of the vector nematodes clearly demonstrates that Riesling resistance is directed against GFLV, the causal agent of grapevine fanleaf

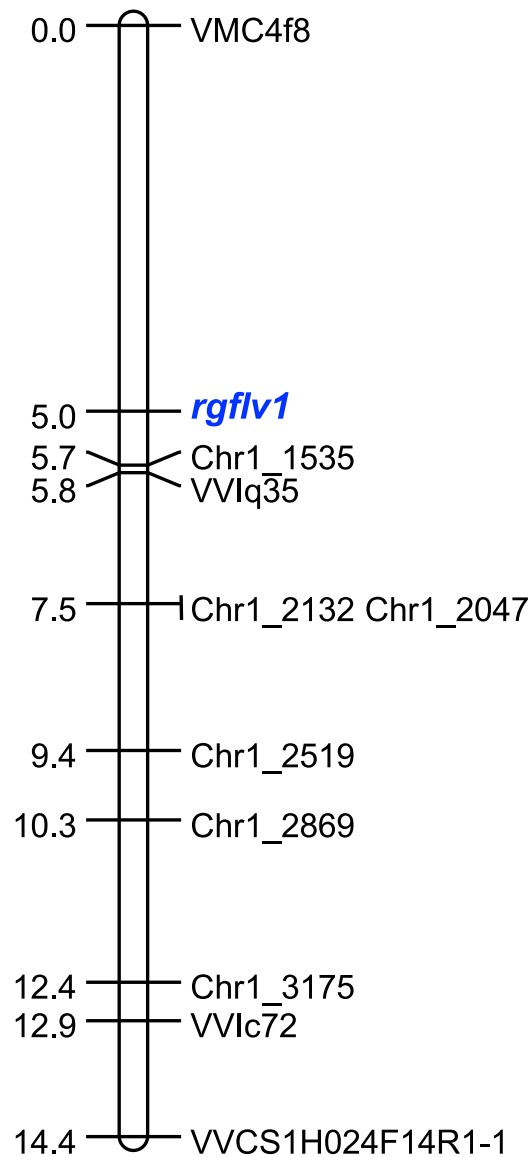

**Fig. 5 Genetic map of the *rgflv1* locus.** Ten SSR markers located on chromosome 1 and surrounding the *rgflv1* locus are represented. Scale on the left is in cM.

disease. Genetically, this resistance is governed by a single recessive factor located on grapevine chromosome 1, which we have named *rgflv1*. These results are of strategic importance given that GFLV is the major viral threat for the wine-growing sector worldwide, leading to vineyard degeneration, and that no sustainable and environmentally friendly method of control exists so far, despite the many efforts by various research groups over decades.

Because viruses have a limited genome size, the viral infection cycle mostly relies on the use of cellular factors, and the completion of this cycle is the result of a complex interplay between virus-encoded and host-encoded factors. In this system, the absence or inadequacy of a single host factor, also called susceptibility factors, led to plants being fully or partially resistant to viruses. Among these factors, components of the eukaryotic translation initiation complex, particularly the eIF4E and eIF4G protein families, were shown to be essential host factors required for RNA virus multiplication and were demonstrated to be highly conserved determinants of plant resistance to viruses[38–40]. Apart from translation, several other functions, including replication,

movement (intracellular, cell-to-cell or long distance) and transmission, may involve plant components recruited by viruses to achieve their infection cycle. Among the ~50 natural recessive resistance loci identified in crop species[35], some were found to be distinct from the *eIF4E* and *eIF4G* families[36]. In barley, protein disulfide isomerase-like 5-1 (PDIL5-1) was identified as the susceptibility factor corresponding to the recessive resistance locus *rym11*, putatively recruited by bymoviruses to act as a cellular chaperone, allowing protein folding or stabilisation, or facilitating transport during virus infection[41]. It was suggested that in *Arabidopsis*, the locus for recessive resistance to watermelon mosaic virus (*rwm1*) encodes an evolutionarily conserved nucleus-encoded chloroplast phosphoglycerate kinase (cPGK2) with a key role in cell metabolism[42]. More recently, in melon, the *cmv1* gene, which confers recessive resistance to cucumber mosaic virus (CMV), was shown to be the vacuolar protein sorting 41 (VPSS41) gene, required for post-Golgi vesicle trafficking towards the vacuole and, thus, putatively used by CMV for its own transport towards the phloem[43].

Five genes identified from the *V. vinifera* reference genome encode proteins of the eIF4E and eIF4G families (Supplementary Table 3). The genes *eIF4E*, *eIF(iso)4E*, *eIF4G*, *eIF(iso)4G1* and *eIF(iso)4G2* are located on chromosomes 10, 5, 15, 4 and 11, respectively. The orthologues of *PDIL5-1*, *cPGK2* and *VPSS41* are located on chromosomes 6, 19 and 6, respectively (Supplementary Table 3). However, none of them is found on chromosome 1. Therefore, *rgflv1* is a putative susceptibility factor, becoming, through a loss of function, a recessive resistance gene against GFLV, distinct from those previously discovered among the natural diversity of plants.

Several previous studies have been conducted with the aim of finding potential sources of resistance to grapevine fanleaf disease in accessions of *V. vinifera* and related species[26,32–34]. These efforts have mainly targeted dagger nematode resistance and have led to the release of several rootstock varieties initially described as resistant to *X. index* feeding[31,44]. However, these rootstocks proved ineffective, since scions grafted on them became infected by the virus a few years after planting[30,31]. Moreover, the complexity of the genetic basis underlying their resistance was a hindrance to their efficient use in breeding.

In contrast, the most important feature of the protection provided by *rgflv1* is its monogenic determinism, which makes it stable through transmission to offspring and easy to use in breeding programmes, despite being recessive. The results of our work represent, to date, the first hope of an effective solution to control fanleaf disease in an environmentally friendly way, which was hitherto an unthinkable prospect. In practical terms, our work paves the way for the development of new GFLV-resistant grapevine rootstocks. This development will be greatly facilitated by marker-assisted selection, already widely used in grapevines[45], thanks to the knowledge of the location of *rgflv1* and its flanking markers. Indeed, the expectations of the grapevine industry are particularly high concerning the control of grapevine fanleaf disease because of the very large decreases in production and economic losses it causes. These GFLV-resistant rootstocks will be a credible alternative to nematicide phytosanitary treatments, which have been banned because of their acute toxicity. In addition, the industry could benefit from the combination of *rgflv1* and resistance to the vector nematode, *X. index*, already described in muscadine grapes[26,27]. These findings will have an impact worldwide, because most vineyards are currently affected by grapevine fanleaf disease. Given the biological characteristics of nematode/*Nepovirus* associations and, notably, the longevity of viruliferous *Xiphinema* species in the absence of any host plant[9], it appears that only a genetic approach could provide effective and durable resistance against nematode-transmitted viral diseases.

More broadly, the discovery of *rgflv1* will contribute to the development of an innovative vine growing system based on the extensive use of resistant varieties with no pesticide input, thus making the vine industry more competitive and sustainable. Of course, resistance to grapevine fanleaf disease should be associated in rootstocks to resistance to phylloxera, which is a soil-borne pest widespread in vineyards. However, conventional breeding (i.e., crossing) can also be used to combine GFLV resistance with resistance to other major diseases, such as downy and powdery mildew, and traits of adaptation to climate change to design a new ideotype of multi-resistant grape varieties; the current context, in terms of regulations, the evolution of society and the concerns of the grapevine industry is particularly favourable to the deployment of these varieties. Above all, new varieties created through conventional breeding do not suffer from the negative societal perceptions generally expressed by consumers towards genetically modified (GM) products, which strongly limits the development of GM-derived resistant material even when a putative resistance to GFLV is identified[23].

To conclude, *rgflv1* appears to be a new, attractive resource, not only for identifying the gene impeding the GFLV life cycle in *Vitis* species, but also for discovering a new susceptibility factor and recessive resistance mechanisms possibly conserved in many crop species. Beyond the interest of this study to provide a solution for a serious and worrying agricultural issue, the characterisation of the genes underlying the *rgflv1* locus could provide important basic information on plant–virus interactions.

## Methods

**Plant material.** Eleven progenies derived from self-fertilisation (S1) of *V. vinifera* varieties or *Vitis* ssp. were used to explore the presence of resistance to GFLV based on recessive determinism (Table 1). A set of five genotypes was used to directly confirm the resistance previously observed in S1 Riesling progeny (Table 2). To determine the target, i.e., the virus or vector nematode, against which the fanleaf resistance observed in Riesling is directed, we compared the ability of *X. index* to reproduce in the Riesling cultivar and two reference genotypes: SO4, a nematode-susceptible genotype and VRH8771, a nematode-resistant genotype. To decipher the genetic determinism of the resistance discovered in Riesling, we first used 14 individuals from a cross between Riesling clone 49 and Gewurztraminer clone 643 (Rs × Gw; Supplementary Table 1)[46]. Then, we developed a new progeny (composed of 113 individuals and called 44628) from the self-fertilisation (S1) of Rs × Gw individual 0071E (Fig. 3).

The plant material was produced either from cuttings, which produces biological replicates, or by sowing seeds. All the S1 plants were produced directly from sowing. The other grapevine genotypes (GFLV-free or GFLV-infected) were produced from green cuttings cultivated on rockwool (Grodan, Roermond, Netherlands) in a climatic chamber with a controlled temperature (24 °C) and light conditions (16 h/8 h photoperiod). After 4–5 weeks, the rooted grapevine plants were transferred to sandy soil and used for *X. index* rearings, GFLV nematode transmission or nematode multiplication assays in a greenhouse. Plants of the S1 and Rs × Gw populations were prepared in the greenhouse conditions described above, and then transferred into large containers filled with vineyard soil naturally infested by viruliferous nematodes.

### Phenotyping

*Double-antibody sandwich enzyme-linked immunosorbent assay.* Plant material was ground at a 1:5 w:v ratio in 0.2 M tris-HCl buffer (pH 8.2) containing 0.8% NaCl, 2% PVP and 0.5% Tween 20 (ref. [47]) and clarified by 5 min of centrifugation at $4000 \times g$. GFLV presence was detected using a commercial kit (Bioreba, Reinach, Switzerland), following the manufacturer's instructions. The absorbance of the hydrolysed substrate (*para*-nitrophenylphosphate) was recorded after 2 h at 405 nm ($OD_{405 nm}$) with a Titertek Multiscan MCC/340 reader (Labsystems, Helsinki, Finland). A healthy plant extract was included in two wells of each 96-well plate as a negative control. For each experiment, a threshold value, above which a tested sample is considered to be positive, was defined as two times the mean of the healthy controls recorded over the whole experiment.

*Production of viruliferous nematodes.* Fig (*Ficus carica*) plants produced from green cuttings (similar to the grapevines) were used to rear non-viruliferous *X. index* in the greenhouse, as described in Villate et al.[24]. Nematodes were extracted from soil samples using the sieving method described by Flegg[48]. An estimated population of 3000 non-viruliferous *X. index* were allowed to feed on four rooted grapevine cuttings infected by the GFLV-F13 isolate in 10-l pots containing loess (1/4) and

sand (3/4, wt/wt). The pots were cultivated in the greenhouse under controlled conditions (22 °C/18 °C day/night with a photoperiod of 16 h/8 h daylight/dark) for at least 2 years to allow the initial nematode population to multiple and acquire GFLV-F13. The presence of GFLV was assessed in groups of 20 *X. index* individuals by reverse-transcription polymerase chain reaction, using primers specific to GFLV-F13 RNA2, as described in Demangeat et al.[49].

*Evaluation of nematode multiplication.* Vines showing homogenous growth were selected, and their roots were exposed to 400 viruliferous *X. index* individuals isolated from the GFLV-F13-infected grapevine rearings. Six replicates were used per grapevine genotype. The plants were cultivated in 2-l containers in the greenhouse under controlled conditions (22 °C/18 °C day/night with a photoperiod of 16 h/8 h daylight/dark) for 12 months, with a 1–2-month dormant period. After 12 months of cultivation, all the soil was recovered from each container, and the plant roots were carefully washed using tap water. *X. index* were extracted from the water suspension based on an adapted Oostenbrink method[50]. The nematodes were counted under a stereomicroscope to determine their number per plant. The ratio of the final number of *X. index* recovered (fn) to the initial number of *X. index* (in)) was used to determine the RF (RF = fn/in) for each plant, with RF = 1 meaning that the tested plant is resistant to the nematode. GFLV presence in the plant roots was assessed by DAS-ELISA.

*GFLV transmission by nematodes in controlled conditions.* The transmission procedure consisted of feeding ca. 300 viruliferous *X. index* for 8 weeks on single grapevines in 0.5 l plastic pots containing loess and sand. The grapevines were cultivated in the greenhouse, as described above. After the feeding period, the grapevines were uprooted, and GFLV transmission was assessed in rootlets by detecting the virus using DAS-ELISA. The grapevines were subsequently transplanted to nematode-free soil and maintained in the greenhouse for at least 2 years, with a 1–2-month dormant period during the winter. DAS-ELISA was then performed on young leaves from newly developed shoots to monitor GFLV infection and subsequent systemic virus spread.

*GFLV transmission by nematodes under semi-natural conditions.* Two-month-old plantlets from cuttings (14 Rs × Gw individuals, with five to six biological replicates per descendant) or seeds (S1 generation from varieties or species and S1 generation from individual 0071E, with one biological replicate per genotype) were transplanted into large containers (5 m³) filled with soil naturally contaminated by viruliferous nematodes, and kept in controlled greenhouse conditions. The soil was transported from naturally infected vineyards. The presence of GFLV was assessed by DAS-ELISA performed on leaves during two to four consecutive years in the spring.

*Resistance phenotyping.* For individuals of population 44628, the highest $OD_{405\,nm}$ value (max_ELISA) observed over the 4 years of the experiment was considered the best indicator of the resistance phenotype. Plants with a max_ELISA value lower than the detection threshold were considered resistant to GFLV, whereas those displaying a value higher than or equal to the threshold were classified as susceptible.

**Genotyping and genetic analysis.** Genomic DNA was extracted from 80 mg of young expanding leaves using a Qiagen DNeasy® 96 Plant kit (Qiagen S.A., Courtaboeuf, France), as described by the supplier. Microsatellite (SSR) analysis was performed as described in Blasi et al.[51].

In the first step, 11 resistant and 11 susceptible individuals of the 44628 progeny were used to identify the chromosomal location of a putative major resistance factor. We screened 101 informative SSR markers, i.e., those previously found to be heterozygous in the 0071E parent and well distributed throughout the genome (Supplementary Fig. 2), from VVS[52], VVMD[53], VrZAG[54], VMC (Vitis Microsatellite Consortium, coordinated by Agrogene, Moissy Cramayel, France), UDV[55] and VVI[56] series, for their ability to detect polymorphisms in both groups of plants. Then, 87 individuals of the 44,628 population were used to precisely map the resistance locus on the targeted chromosome. Six new SSR markers were developed from *V. vinifera*'s reference genome (PN40024 12Xv2; https://urgi.versailles.inra.fr/jbrowse/gmod_jbrowse/?data=myData%2FVitis%2Fdata_gff&loc=chr1%3A9691784.14538491&tracks=Vitis%20vinifera%20cvPN40024%20assembly%2012XV2%2CCRIBI_V1%2CREPET_TE%2Cscaffolds%2CSNP_Discovery_Vitis_vinifera&highlight=; Supplementary Table 2) to densify markers in the region of interest.

Linkage analysis was performed with JoinMap 3.0 (ref. [57]), enabling the analysis of self-pollinated populations derived from a heterozygous parent. The genotypes of the SSR markers were encoded according to an <hkxhk> segregation type. The same segregation type was used for resistance to GFLV, with susceptibility being encoded as dominant (h−) and resistance as recessive (kk). Recombination fractions were converted into centimorgans (cM), using the Kosambi function[58]. The threshold value of the logarithm of odds (LOD) score was set at 4.0 to claim linkage between markers, with the maximum fraction of recombination at 0.45. The goodness of fit between the observed and expected Mendelian ratios was analysed for each marker locus using a $\chi^2$ test.

**Reporting summary**. Further information on research design is available in the Nature Research Reporting Summary linked to this article.

## Data availability
The authors declare that the data supporting the findings of this study are available within the paper and its supplementary information files.

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

## Acknowledgements

We gratefully acknowledge the Fondation Jean Poupelain for funding our work through the HealthyGrape2 programme. Funds were also provided by the European Regional Development Fund (ERDF) in the framework of the INTERREG V Upper Rhine programme Vitifutur, transcending borders with every project, by the Agence Nationale de la Recherche (ANR-08-GEMN-128) and the INRAE Plant Biology and Breeding Department. We are grateful to V. Thareau for providing primer sequences for SSR marker VVCS1H024F14R1-1, to E. Duchêne for providing useful information on the polymorphism of SSR markers in the Rs × Gw progeny and to P. Mestre for helpful discussions that led to the improvement of the manuscript. We thank J. Misbach, L. Ley and the members of the Unité Expérimentale Agronomique et Viticole INRAE-Colmar for excellent assistance in plant maintenance and growth.

## Author contributions

O.L. and D.M. designed the research; D.M. supervised research; S.D., E.P., V.D., G.D., S.G., A.A., C.G. and M.B. performed the research; V.D. developed the plant material; G.D. developed the animal material; D.M. mapped the *rgflv1* locus; S.D., E.P., G.D. and D.M. analysed the data; S.D., G.D. and D.M. wrote the first draft of the manuscript; S.D., E.P., G.D., O.L. and D.M. performed critical revisions of the manuscript.

## Competing interests

The authors declare no competing interests.
