## [Peer Review File · Communications Biology]

Reviewers' Comments:

Reviewer #1:

Remarks to the Author:

This manuscript describes research work performed during many years in the search for resistance to Grapevine Fanleaf virus (GFLV) in *Vitis vinifera*. They study the progeny of 11 varieties of *V. vinifera* to study their level of heterozygosity and focus on the variety Riesling, the only one with a resistant progeny. Then they study the F2 obtained with a susceptible variety and conclude that the resistance is monogenic and recessive. Ulterior mapping finds a region of 1,1 Mb interval at chromosome 1.

This is a sustained effort done for many years, as corresponds for building mapping populations in trees. It has been carried out in a simple but correct methodological way, so that it is convincing, and the conclusions are supported by the results obtained. The results are novel for the field of grapevine and interesting for the grapevine community and for the plant-pathogen community, since it describes a new recessive resistance gene and close molecular markers that could help the introgression in other varieties, minimizing the use of pesticides. The statistical treatment is appropriate and the level of detail given in the Materials and Methods section is enough for a reproduction of the results elsewhere.

There are no major concerns on this paper. In summary, this reviewer thinks that this paper should be published as it is.

Reviewer #2:

Remarks to the Author:

This manuscript aims a better understanding of GFLV-resistance in grapevines. The authors report a new single resistance factor located on grapevine chromosome 1, named *rgflv1*. The results showed in this manuscript are novel and provide potentially compelling evidence on plant resistance for viruses. However, several concerns about the data, the interpretation of the data, and some additional experiments/analyses are needed.

The authors indicate that the Riesling cultivar displays resistance to GFLV determined by a single locus based on its susceptibility to the viral vector X. index and S1 population analysis (cross between Riesling and the susceptible cultivar Gewurztraminer). They analyzed a set of 87 individuals from one generation of self-fertilization of Rs x Gw by resistance phenotyping of GFLV for four years. The authors used the maximal DAS-ELISA value. Nevertheless, they do not indicate the number of assays performed and the correlation between these experiments. The conditions of the resistance/susceptible experiments are usually variable and affect their results. For this reason, the authors could include all experiments and indicate if the distribution and the consequences in the chi-square test (3:1 Mendelian segregation). Another concern is related to the number of individuals analyzed. The authors showed in Figure 3 that F1 (Rs x Gw) was 100% susceptible, but in Table S1, it was 83-100%. The authors only used 14 individuals from F1 population. It is essential to include a chi-square test and more individuals in this analysis. Another concern is related to the number of individuals in the S1 population analysis. For example, Blasi et al., 2011 used 232 individuals from the S1 population studying grapevine downy mildew resistance. The authors include 87 individuals from S1 progeny and of them, only 21 individuals were used for genotyping analysis. More convincing analysis could be performed with the 87 individuals. It is not clear if the marker segregation in *rgflv1* locus on chromosome 1 was present in parents genotypes and other varieties included in this work. In addition, the authors do not indicate if the GFLV resistant genotypes showed susceptibility to X. index like the Riesling genotype.

Dear Colleague,

Please find enclosed a revised version of our article entitled “A single resistance factor to solve vineyard degeneration due to grapevine fanleaf virus” by Samia Djennane et al.

All our revisions are marked in the main text file using the track changes mode. The revisions in supplementary information and supplementary tables are written in red.

Reviewer #1 having considered that our article should be published as it is, here are our point-to-point answers to the comments of reviewer #2 which we have numbered as follows.

1. *The authors indicate that the Riesling cultivar displays resistance to GFLV determined by a single locus based on its susceptibility to the viral vector X. index and S1 population analysis (cross between Riesling and the susceptible cultivar Gewurztraminer). They analyzed a set of 87 individuals from one generation of self-fertilization of Rs x Gw by resistance phenotyping of GFLV for four years. The authors used the maximal DAS-ELISA value. Nevertheless, they do not indicate the number of assays performed and the correlation between these experiments. The conditions of the resistance/susceptible experiments are usually variable and affect their results. For this reason, the authors could include all experiments and indicate if the distribution and the consequences in the chi-square test (3:1 Mendelian segregation).*

The individuals of the 44628 S1 segregating population were studied altogether in a single experimental design and the experiment lasted four years. For each individual, the presence of GFLV was assessed annually by DAS-ELISA performed on leaves in the spring. We considered the highest DAS-ELISA value (called max_ELISA) observed over the four years of the experiment as the most

reliable indicator of the resistance phenotype. These details are given in the 'Results' (lines 130-145) and 'Materials and methods' (lines 334-342) sections.

This approach was already used in similar cases (Zhang et al. Theor Appl Genet (2009) 119:1039–1051; Rubio et al. BMC Plant Biology (2020) 20:213), i.e. when the experimental system involved a plant-animal interaction and, because of the characteristics of its biology, undergoes variations that can hardly be avoided. In our case, a punctually low DAS-ELISA value may result from delayed inoculation and/or heterogeneity in the distribution of the virus in the plant. In addition, given the number of descendants required for genetic analysis and the cumbersome nature of the resistance phenotyping system, it was not possible to test several biological replicates of each offspring. This is the reason why we chose to phenotype our population with one biological replicate per genotype but through a multi-year inoculation.

To meet the request of Reviewer #2, we have added in Table S2 the detail of annual results recorded on each plant over four years and explained in the legend the decision rule used to validate the acquired data.

2. Another concern is related to the number of individuals analyzed. The authors showed in Figure 3 that F1 (Rs x Gw) was 100% susceptible, but in Table S1, it was 83-100%.

There is no contradiction between the figures in Figure 3 and those in Table S1. Table S1 represents the analysis of 14 descendants of a Riesling x Gewurztraminer cross. For each descendant, 5 to 7 cuttings obtained by vegetative propagation (i.e. biological replicates) were analyzed after infection. Table S1 represents the observed infection rate (number of infected cuttings /number of assayed cuttings) for each of the descendants. Most of them show 100% of infected cuttings, except one with an infection rate of 83%. Nevertheless, all the descendants are susceptible. This is why we indicate 100% susceptible in Figure 3. The experimental details are given in the Results section (lines 122-128) and in the legend of Table S1.

To make the reading of these results clearer, we have added to Table S1 a column indicating the status of each descendant and have improved the table legend by adding details of the protocol.

3. The authors only used 14 individuals from F1 population. It is essential to include a chi-square test and more individuals in this analysis.

The experiment cited by Reviewer #2 shows that the Riesling resistance to GFLV is not transmitted to offspring when crossed with the susceptible parent Gewurztraminer. The most parsimonious hypothesis to explain this observation is that resistance is recessive.

To carry out a chi-square test, it is necessary to compare the observed data distribution with a theoretical distribution. In our case, if we compare the observed distribution with the theoretical distribution obtained under the hypothesis of recessive resistance, p-value will be close to 1, since 100% of observed individuals are susceptible. Rather than increasing the number of Riesling x Gewurztraminer offspring tested, we chose to test our hypothesis with a new progeny.

This is clearly mentioned in the 'Results' (line 130) section. The following experiments totally confirm that the recessivity hypothesis is correct and relevant.

4. Another concern is related to the number of individuals in the S1 population analysis. For example, Blasi et al., 2011 used 232 individuals from the S1 population studying grapevine downy mildew resistance. The authors include 87 individuals from S1 progeny and of them, only 21 individuals were used for genotyping analysis. More convincing analysis could be performed with the 87 individuals.

In Blasi et al (2011), the study focused on quantitative resistance to downy mildew. This work differs from our own in two essential ways which led us to choose the experimental design we used.

First of all, downy mildew resistance can be assessed in less than 7 days on leaf discs through a bioassay occupying a few square metres in the greenhouse, whereas in our study, resistance assessment requires large containers filled with 5 cubic metres of infested soil, for 4 years. Secondly, the downy mildew study aimed to investigate the genetic determinants of quantitative resistance, whereas our study reveals a qualitative genetic factor.

As explained in the 'Results' (lines 152-172) and 'Materials and methods' (lines 349-358) sections, we proceeded in two steps. Briefly, in the first step, 22 individuals (11 resistant and 11 susceptible) of the segregating progeny were used to identify the chromosomal location of a putative major resistance factor thanks to molecular markers well distributed throughout the genome. Then, the 87 individuals (all those that could be phenotyped) were used to precisely map the resistance locus on the targeted chromosome. The strategy used in the first step is derived from 'bulk segregant analysis developed by Michelmore et al (PNAS (1991), 88:9828-9832) and, since, widely used for rapidly identifying markers linked to any specific gene or genomic region (to date 1,968 references in Web of Science).

The 'Results' (line 162) and 'Materials and methods' (lines 357-358) sections have been amended to make clearer the numbers of individuals used at each step.

*5. It is not clear if the marker segregation in *rgflv1* locus on chromosome 1 was present in parents genotypes and other varieties included in this work.*

The allele found in the mapping population were present in the parent (0071) and the grandparents, Riesling and Gewurztraminer.

The 0071, Riesling and Gewurztraminer genotypes were added in Table S3 for all tested markers.

6. In addition, the authors do not indicate if the GFLV resistant genotypes showed susceptibility to X. index like the Riesling genotype.

Once it was established that the resistance of the Riesling was directed against the GFLV and not against its vector, X. index, we focused on the GFLV resistance and no longer analysed the X. index resistance. In the context of our work, there is no reason to believe that offsprings from X. index-susceptible individuals are resistant to X. index.

Best regards,

Dr. Didier Merdinoglu

Corresponding author

didier.merdinoglu@inrae.fr